# The direct and indirect costs of cardiovascular diseases in Brazil

**Jevuks Matheus de Araújo** [1], **Rômulo Eufrosino de Alencar Rodrigues** [1], **Adélia da Costa Pereira de Arruda Neta** [2], **Flávia Emília Leite Lima Ferreira** [2‡], **Rafaela Lira Formiga Cavalcanti de Lima** [2‡], **Rodrigo Pinheiro de Toledo Vianna** [2‡], **Lucas Vasconcelos Leitão Moreira** [3‡], **José Moreira da Silva Neto** [4‡], **Patrícia Vasconcelos Leitão Moreira** [2] *

1 Department of Economy, Federal University of Paraiba, João Pessoa, Paraíba, Brazil, 2 Department of Nutrition, Federal University of Paraiba, João Pessoa, Paraíba, Brazil, 3 Department of Medicine, UNIPÊ, João Pessoa, Paraíba, Brazil, 4 Technical Health School, Federal University of Paraíba, João Pessoa, Paraíba, Brazil

☉ These authors contributed equally to this work.
‡ FELLF, RLFCL, RPTV, LVLM and JMSN also contributed equally to this work.
* patricia.moreira@academico.ufpb.br, patriciamoreira1111@hotmail.com

**Data Availability Statement:** The microdata used in the study are available in the following public domain and open access databases: - Brazilian

## Abstract

### Objective

To evaluate the direct and indirect costs of cardiovascular diseases (such as coronary heart disease and stroke) by sex and age group, attributed to the excessive consumption of salt, saturated fat and trans fat in Brazil.

### Materials and methods

The data for estimating the Population Attributable Fraction (PAF) corresponding to the consumption of salt, saturated fat and trans-fat were obtained from the Household Budget Survey 2017–2018. The calculation of direct costs for cardiovascular diseases (CVD) was made from the accounting sum of costs with hospitalizations and outpatient care found in the National Health System (Hospital Information System and Outpatient Information System), from 2017 to 2019, including the costs of treatment, such as medical consultations, medical procedures, and drugs. Regarding the indirect costs, they were measured by the loss of human capital, given the premature death, resulting in loss of productivity. To define the attributable costs, they were multiplied by the PAF.

### Results

Higher burden of CVD attributable to the consumption of salt, saturated fat and trans fat were observed in younger individuals, which progressively decreased with advancing age, but still generated economic costs in the order of US$ 7.18 billion, in addition to 1.53 million productive years of life lost (YLL) to premature death, if considering salt as an inducer. Although attributable burden of CVD is higher among younger individuals, the highest costs are associated with males aged 45 to 74 years old for direct costs and 45 to 64 years old for indirect costs.

Institute of Geography and Statistics: (https://www.ibge.gov.br/estatisticas/sociais/saude/24786-pesquisa-de-orcamentos-familiares-2.html?=&t=microdados); (https://www.ibge.gov.br/estatisticas/sociais/populacao/9127-pesquisa-nacional-por-amostra-de-domicilios.html?=&t=microdados); and (https://www.ibge.gov.br/estatisticas/sociais/populacao/9126-tabuas-completas-de-mortalidade.html?=&t=downloads) - Informatics Department of the Brazilian Unified Health System (DATASUS) for the Outpatient Information System (SIA, specifically, SIA-PA [production subsystem]), Hospital Information System (SIH, specifically, SIA-RD [subsystem of accepted admissions]) and Mortality Information System (SIM, specifically, SIM-DO [death certificate subsystem], all on the same link for access to raw data (https://datasus.saude.gov.br/transferencia-de-arquivos/).

**Funding:** The National Council for Scientific and Technological Development (Conselho Nacional de Desenvolvimento Científico e Tecnológico – CNPq) supported all the funding received during this study; Grant number 442891/2019-9. The funders had no role in study design, data collection and analysis, decision to publish, or preparation of the manuscript.

**Competing interests:** The authors have declared that no competing interests exist.

## Conclusion

The attributable fractions to consumption of salt are the ones that cause the most effects on CVD, followed by saturated fat and trans fat, with direct and indirect costs being higher for males.

## Introduction

Cardiovascular diseases (CVD) are one of the key challenges in health globally, since they are the main cause of premature deaths and compromised productivity, reaching a mortality rate in 2019 of around 32% in the total population and 38% in individuals under the age of 70 [1–3]. Three quarters of the total of these deaths are concentrated in low- and middle-income countries [3], where the population is more vulnerable to the risk factors of these diseases. Furthermore, with the increase in longevity and the considerable reduction of infectious diseases, the aging of the population also contributes to increase the number of CVD [4, 5].

In 2018, the total population of Brazil was 208,494,900 people, of whom 51.08% were women and 69.43% were 15 to 64 years old. Life expectancy at birth was 72.74 years for men and 79.80 years for women [6]. Specifically in Brazil, between 2000 and 2018, crude mortality rates from CVD have been decreasing in adults over 25 years of age, of both sexes, except in men over 85 years of age [7]. Even in this scenario, CVD were responsible for 28% of all deaths that occurred between 2010 and 2015, and 38% of this number occurred in the productive age group (18 to 65 years) [8].

In addition to the irreversible social losses in the family environment, CVD have a considerable weight in public and private financial costs (direct costs) due to hospitalizations, monitoring, treatment, and others, and in the loss of productivity (indirect costs). Health financing in Brazil comes from public and private sources. The model covers the Brazilian National Health System (SUS—*Sistema Único de Saúde*), supported by taxes and contributions collected at the federal, state and municipal levels, and the Complementary Health System, with resources from companies and individuals, with 71% of the Brazilian population using this system [9].

In Brazil, estimated costs for CVD were around R$37.1 billion in 2015 [7]. Of these estimated CVD costs, 61% were estimated for premature death from CVD, direct costs for hospitalizations and consultations were 22%, and disease-related lost productivity costs were 15% of the total [8]. Therefore, the growth of these recurrent costs represents an important problem for health systems, as well as for their socioeconomic impacts [10–12].

Nowadays, Brazilians have been going through gradual changes in eating patterns, evidently perceived on the excessive consumption of nutrients linked to the causes of Non communicable diseases (NCD), such as sodium, sugars, and fats [13]. An increase in the consumption of ultra-processed foods (UPF) by the Brazilian population has been observed [13, 14]. These UPF have high levels of the above mentioned nutrients, with scientific evidence of their relation to obesity [15, 16], diabetes mellitus [17, 18], hypertension [19] and cardiovascular diseases [20–22].

Few studies have been carried out with the objective of attributing the effect of nutrient consumption to the cause of CVD and its consequent costs [23, 24]. In Brazil, it was estimated the attributable burden of excessive salt consumption in cardiovascular diseases, showed savings of 9.4% of total hospital costs for CVD, if the average salt intake of Brazilians were reduced to 5 g/d [23]. In this sense, studies that seek to minimize the socioeconomic impacts of this

nature can contribute to stimulating the improvement and development of public policies to combat and prevent these diseases.

The Population Attributable Fraction (PAF) is a measure of public health impact that has been widely used by the World Health Organization, based on Global Burden Disease (GBD) data, in order to determine goals, prioritize interventions and build public policies [25, 26]. In addition, the PAF has also been used to provide information on economic costs attributable to some risk factors, such as salt consumption [27].

There are two ways to estimate health costs for a disease. The first is "top-down", going from the total values at the national level of the set of all diseases and, through a disaggregation process, arriving at the level at which the cost of the disease under analysis is found. The second is "bottom-up", and through this method, estimates are made for a sample of cases and are extrapolated to the total number of individuals [28].

In Brazil's context, it is possible to obtain the total direct costs related to a given pathology in the National Health System, which can be disaggregated by level of health care (outpatient and hospital), sex and age groups. Thus, the best approach to be used in Brazil is the top-down approach, from the perspective of public health services, based on health cost data, available in the information systems of the Ministry of Health [28].

Thereby, this study assessed the direct and indirect costs of coronary heart disease (CHD) and stroke by sex and age group, concentrating the etiology of the CVD to the excessive consumption of salt, saturated fat, and trans fat, between 2017 and 2019, in Brazil.

## Materials and methods

### Data

Several databases were referred to collect and analyse CVD cost related information between 2017 and 2019 for Brazil. For the estimation of the Population Attributable Fraction corresponding to salt, saturated fat and trans fat, data were obtained from the Household Budget Survey (HBS) (*Pesquisa de Orçamento Familiar*—POF) 2017/2018, from the Brazilian Institute of Geography and Statistics, which provides information on household food consumption [14, 29]. Data for direct costs were focused on expenses with hospital admissions, from the Hospital Information System (*Sistema de Informação Hospitalar*, SIH/SUS) [30], and outpatient care, from the Outpatient Information System (*Sistema de Informação Ambulatorial*, SIA/SUS) [30] (including the costs of treatment, such as medical consultations, medical procedures, and drugs), both covering the entire proposed period, collected from the Brazilian National Health System (SUS—*Sistema Único de Saúde*) database (DATASUS) [30], Ministry of Health of Brazil. DATASUS is an open public domain secondary database that includes a wide range of systems and subsystems that contain time-varying information between them, and a variety of information systems on Brazilians in relation to mortality, hospital care, outpatient care, basic health care, live births, physical structure of hospitals and clinics, professionals included in the SUS, notifications of contamination of epidemiological diseases, among others [31].

Regarding indirect costs, three databases were necessary: the observational characteristics of deceased individuals which were attained from the DATASUS Mortality Information System (SIM, *Sistema de Informação de Mortalidade*) between 2017 and 2019 [32]; for the characteristics and income of living individuals, data were collected from the Brazilian Institute of Geography and Statistics (IBGE) in the National Household Sample Survey (NHSS), for the year 2015 [33]; finally, the mortality probabilities of the population by age and sex were obtained from the IBGE Complete Mortality Tables (2017, 2018, 2019) [32].

Moreover, in order to filter data on CVD that led to hospitalizations, outpatient care and mortality, these diseases were classified according to the 10th review of the International

Classification of Diseases—ICD 10, with the codes I20-I25 for CHD, and I60-I69 for stroke [34]. The age groups were formed following the standard of the Pan American Health Organization for both sexes: 25–34 years old, 35–44 years old, 55–64 years old, 65–74 years old, 75–84 years old and > 85 years old (for indirect costs, the last age evaluated was 65 years).

In addition, all monetary values were corrected using the Index National Price on Expanded Consumers (Índice Nacional de Preços ao Consumidor Amplo, IPCA/IBGE) in Brazilian currency (R$—*reais*) for 2019 and, subsequently, converted to US dollars ($) at the average exchange rate for 2019, corresponding to 3.944 R$/U$S (0.25345 U$S/R$), made available by the Institute of Applied Economic Research (IPEA) [35].

The research protocol was approved by The Ethics Committee of Federal University of Paraíba with consent number 3.843.739. As these are secondary data made available on public domain sites of the Brazilian Unified Health System, ethics committee waived the requirement for informed consent.

## Population Attributable Fraction (PAF)

The proportion of the risk of developing CVD that would be reduced in each period if the consumption of salt, saturated fat and trans fat were reduced to their ideal consumption (TMREL), is called Population Attributable Fraction (PAF), which was defined as:

$$PAF = \frac{\int_0^l RR(x)P(x)d(x) - RR(x)TMREL}{\int_0^l RR(x)P(x)d(x)} \tag{1}$$

Where P(x) is the distribution of current food consumption, RR(x) is the relative risk of developing CVD at exposure level (x), and (l) is the maximum level of exposure, that is, of consumption of the nutrient. The TMREL is the theoretical minimum risk level, that is the consumption level considered ideal to minimize the risk at the population level. In this study, the TMREL for salt was 5g/day, for saturated fat was 10% of the total daily energy value and for trans fat was 1% of the total daily energy value [10].

O RR(x) is defined to be:

$$\exp\left(\beta(x - y(x))\right) \text{ if } x - y(x) \geq 0 \tag{2}$$

or

$$1 \text{ if } x - y(x) < 0$$

where β is the change in log relative risk per unit of exposure [10, 36], x is the current exposure level, and y(x) is the TMREL.

Food consumption was measured using food records, applied on non-consecutive days, in which individuals were instructed to record and report in detail the names of the foods consumed, the type of preparation, the measure used, the amount consumed, the time and whether the consumption of the food occurred at home or outside the home, with their serving sizes converted from standard units or household measures, to grams, using a common reference table to HBS [14]. A second measure of food records was performed in a sub-sample of HBS [14]. Habitual nutrient intake was estimated using the Multiple Source Method (MSM). This last method is suitable to estimate the usual individual intake for repeated measurements and a defined period [37].

To verify the distribution that best fits the sample data, the adherence test was performed, which confirmed that the salt consumption data showed a log-normal distribution, while the saturated and trans fat data showed a gamma distribution. Both distributions were used in the PAF calculation.

## Direct cost

The total direct costs (TDC) of the CVD were attained by the sum of the costs of the admissions and outpatient care in the National Health System (SUS). Hence, public expenses for CHD and stroke were estimated in both, the Hospital Information System (*Sistema de Informação Hospitalar*, SIH/SUS) and the Ambulatory Information System (*Sistema de Informação Ambulatorial*, SIA/SUS). The base equation is presented below:

$$TDC_{t,a,b} = \sum_{n=1}^{n} SIH_{ij,t,a} + \sum_{k=1}^{k} SIA_{aj,t,a} \qquad (3)$$

Where, $TDC_{t,a}$ is the total direct cost of all *n* hospital admissions, $SIH_{ij,t,a}$, plus *k* outpatient care, $SIA_{aj,t,a}$, of individuals (ij) and (aj), in the age group (a), sex (b) at time (t), for CVD diseases (CHD and stroke). To isolate the effect attributable to excessive nutrient intake, we weight Eq 3 by the PAF:

$$TCD_{s,st,tf,t,a,b} = (\sum_{n=1}^{n} SIH_{ij,t,a,b} + \sum_{k=1}^{k} SIA_{aj,t,a,b}) * PAF_{s,sf,tf} \qquad (4)$$

Therefore, $PAF_{s,sf,tf,a,b}$ represents the population attributable fraction by age group and sex, with respect to salt (s), saturated fat (sf), trans-fat (tf), that is, the portion attributable to the costs of excess consumption of these nutrients.

## Indirect cost

The main approaches in indirect costs analysis are human capital that is associated with lost productivity and friction costs that estimates the costs of worker replacement. We used the human capital approach and estimated the productivity losses based on wages over the working life of the worker. Observational variables are used that characterize the income of living individuals, correlating with the observational characteristics of individuals who had deaths from CVD under the age of 65 years. Thus, this combination provides the income of the individual who died given the probability of being alive.

The method used in Ywata *et al*. (2008) [38] was partially followed, where they measured the loss of human capital caused by premature external deaths, such as those caused by homicide and traffic accidents. Using the income data of individuals and crossing these data with information regarding age group, sex, schooling, and geographic location of the residence, it is possible to establish an estimate of the average income for subgroups of the population [33]. To obtain the income stream of these population subgroups we project the estimated income over time. Thus, to understand the loss of productivity, the reduction of income generated by the deaths caused by the CHD and stroke imputed to the subgroups of the dead population the estimated income stream for the same subgroups of the population that maintained their work activities. The estimates of average income and projection of future income follow the methodology described in Ywata et al [38]. To separate the causes, we weighed the income stream of each population subgroup by the PAF.

In this study, irreversible productivity losses due to the premature death of individuals aged 25 to 65 years caused by CVD were considered. For this task, six explanatory variables for income that are available in the NHSS, and SIM databases were considered: state of the country in which the individual resides (residence), age, sex, education level, color/race, and marital status [39].

Since it is impossible to accurately define the future income of deceased people, data from living people contained in the NHSS database, were used to pair the dead individuals with the

observational variables from both databases. In other words, the income of individuals contained in the NHSS was used for the dead individuals of the SIM database, looking to match the six selected variables, since the income is found only in the NHSS database. The observations perfectly matched between the bases in 83.8%. For the imperfectly matched combinations, variables were being removed according to their importance for the explanation of income. First, marital status and then, color/race. As it is not possible to define whether the level of education, residence status and marital status of deceased individuals would change over time, only the age was increased. Thus, an individual who died at 25 years of age, in 2019, had his age extended to the following years to 26, 27. . ., until 65 years of age, in 2049, which is the maximum age of productivity considered, given the probability of survival. Therefore, all other variables values are constant in time. To bring the future income stream to the present, we applied the net present value (NPV) which allows to measure the value of money in time:

$$\text{TIC}_t, i = \sum_{X=Di}^{T} \frac{1}{(1+d)^{(x-Di)}} * \Pr(Fi > x | Fi \geq Di) * Wi \quad (5)$$

In the occasion that (d) is the annual discount rate of 3%, ($W_i$) is the expected annual income of the individual (*i*) present in the SIM. 'T' is the maximum productive age, in this case, 65 years. The probability Pr *(Fi > x|Fi ≥ Di)*, is the probability that the individual is alive at age (*x*), given that he did not die at the age of (*Di*), that is, the age registered in the SIM. The age of death is indicated by ($F_i$), given that individual (i) has already died. Finally, $TIC_{t,i}$ is the total indirect cost of individual (i) at time (t). Similarly, to the direct costs, to isolate the effect of nutrients on indirect costs, it is:

$$\text{TIC}_t, i = \sum_{X=Di}^{T} \frac{1}{(1+d)^{(x-Di)}} * \Pr(Fi > x | Fi \geq Di) * Wi * PAFs, sf, tf \quad (6)$$

Hence, for the general total indirect cost and for the general total attributable indirect cost, they are attained by summing up the individual costs and grouping them by age group. In addition, if the individual had not died, the expected years of productive life lost due to premature deaths were calculated accordingly:

$$\text{YLL}_t, i, s, sf, tf = \sum_{x=Di}^{T} x * \Pr(Fi > x | Fi \geq Di) * PAFs, st, tf \quad (7)$$

In the occasion that $YLL_{t,i,s,sf,tf}$ are the Years of Life Lost (YLL) by the individual due to premature death from CVD attributable to excessive use of salt (s), saturated fat (sf) and trans fat (tf).

This methodological approach to the estimation of indirect costs was considered due to the costs associated with permanent loss of productivity given the premature death from CVD, caused by excessive consumption of nutrients, being, in hypothesis, the highest within this cost modality [38]. In addition, the estimates are more realistic, as real observational variables are used as determinants of the income of deceased individuals, reducing possible bias in the results [38].

## Results

Table 1 brings the estimates for the impacts of the excessive consumption of salt, saturated fat and trans-fat by age group and sex. The attributable fractions to salt are the ones that cause the most effects on CVD, followed by saturated fat and trans fat. Among the sexes, males have a greater share related to the excessive consumption of all nutrients associated to CVD, with salt being responsible for the highest value, 99.7% in the 25–34 age group. Likewise, for females,

**Table 1. Population Attributable Fraction (PAF) to salt, saturated fat and trans fat by age and sex.**

| Age group (Years) | Male | | | Female | | |
|---|---|---|---|---|---|---|
| | PAFs* | PAFsf** | PAFtf*** | PAFs* | PAFsf** | PAFtf*** |
| 25–34 | 0.997 | 0.749 | 0.163 | 0.858 | 0.627 | 0.143 |
| 35–44 | 0.953 | 0.687 | 0.089 | 0.690 | 0.521 | 0.071 |
| 45–54 | 0.862 | 0.574 | 0.057 | 0.518 | 0.519 | 0.064 |
| 55–64 | 0.722 | 0.361 | 0.020 | 0.374 | 0.329 | 0.026 |
| 65–74 | 0.485 | 0.266 | 0.013 | 0.241 | 0.227 | 0.016 |
| 75–84 | 0.343 | 0.156 | 0.003 | 0.153 | 0.155 | 0.013 |
| 85 > = | 0.145 | 0.093 | 0.005 | 0.068 | 0.116 | 0.006 |

PAFs: Population Attributable Fraction to salt; PAFsf: Population Attributable Fraction to saturated fat; PAFtf: Population Attributable Fraction to trans fat

the highest values for nutrients are all in the 25–34 age group: 85.8% for salt, 62.7% for saturated fat and 14.4% for trans fat. Between the sexes, the greatest similarity is found in the PAFtf, with all PAF having an inverse relationship with the age.

Moreover, it is significant to highlight that there is an intersection between the consumption of nutrients by the individuals. This means that the individual who consumes salt excessively, also consumes fats, so the sum of PAF exceeds in most cases.

Regarding direct costs, between 2017 and 2019, there were approximately 1.5 million admissions and 21 million visits by Brazilian clinics caused by CVD, most predominantly by males in the 55–64 age group. In this period, hospitalizations and outpatient care resulting from excessive consumption of salt demanded US$ 730.91 million for the public resources, with approximately 77.5% of expenses for males, as shown in Table 2. We observed, for females, that the highest amounts of hospitalizations and outpatient visits were in the 65–74 years age group. Also, the direct costs attributable to salt, saturated fat and trans-fat of, were respectively: US$ 40.44, US$ 38.07 and US$ 2.62 million. Between 2017 and 2019, total direct costs for males increased by 2.1% and for females the growth was 2.6%.

Furthermore, Table 3 provides the data regarding the indirect costs as consequence of the premature deaths from CVD associated to the excessive intake of salt, saturated fat and trans fat. During the studied period of time, a total of 855,370 deaths from CVD were registered in Brazil, of which 62.17% were males, with an average age of 54 years of age. Deaths resulting from excessive salt intake resulted in costs of US$ 6.45 billion to the economy due to premature loss of productivity, to which males accounted for about 80.5% of this value. Also, the age group that presents the highest indirect costs for both sexes and for all nutrients are people from 45 to 54 years old. However, the male participation in these costs is increasing for all years and all nutrients.

Finally, an alternative way to quantify productivity losses that generate indirect costs, is to estimate the expected productive years of life lost due to premature death, as shown in Table 4. Interestingly, as a result of the 855,370 premature deaths between 2017 and 2019, a total of 1.53 million of potential productive years of these individuals were lost (about 72.5% for males), considering the excessive use of salt; for saturated fats a total of 1.09 million years (about 66.6% for males) and in relation to trans fats, a total of 132.41 thousand years (about 63.5% for men), having the 45–54 age group with the highest aggregate loss.

## Discussion

Study showed that the direct and indirect costs of CVD (such as CHD and stroke) attributed to the consumption of salt, saturated fat and trans fats, by sex and age group, between 2017

**Table 2. Direct costs attributable to excess consumption of nutrients (salt, saturated fat and trans-fat) by sex and age group in US$ per Million.**

| Sex | Age | 2017 | | | 2018 | | | 2019 | | |
|---|---|---|---|---|---|---|---|---|---|---|
| | | Salt | Sat. Fat* | Trans fat | Salt | Sat. Fat* | Trans fat | Salt | Sat. Fat* | Trans fat |
| **Male (M)** | 25–34 | 3.56 | 2.67 | 0.58 | 3.54 | 2.66 | 0.58 | 3.41 | 2.56 | 0.56 |
| | 35–44 | 12.85 | 9.26 | 1.20 | 12.89 | 9.30 | 1.21 | 13.16 | 9.49 | 1.23 |
| | 45–54 | 42.33 | 28.19 | 2.79 | 40.34 | 26.87 | 2.66 | 40.74 | 27.13 | 2.69 |
| | 55–64 | 66.66 | 33.29 | 1.80 | 66.51 | 33.22 | 1.80 | 68.43 | 34.18 | 1.85 |
| | 65–74 | 39.53 | 21.67 | 1.10 | 40.86 | 22.40 | 1.13 | 41.68 | 22.85 | 1.16 |
| | 75–84 | 12.40 | 5.64 | 0.11 | 12.66 | 5.76 | 0.11 | 13.24 | 6.03 | 0.12 |
| | 85 > = | 0.97 | 0.62 | 0.03 | 0.92 | 0.59 | 0.03 | 1.00 | 0.64 | 0.03 |
| | Total | 178.29 | 101.35 | 7.62 | 177.73 | 100.79 | 7.52 | 181.67 | 102.88 | 7.63 |
| | **Age** | **2017** | | | **2018** | | | **2019** | | |
| | | Salt | Sat. Fat* | Trans fat | Salt | Sat. Fat* | Trans fat | Salt | Sat. Fat* | Trans fat |
| **Female (F)** | 25–34 | 2.43 | 1.78 | 0.40 | 2.29 | 1.67 | 0.38 | 2.28 | 1.67 | 0.38 |
| | 35–44 | 7.17 | 5.41 | 0.74 | 7.06 | 5.32 | 0.73 | 7.39 | 5.57 | 0.76 |
| | 45–54 | 15.99 | 16.02 | 1.99 | 15.43 | 15.45 | 1.92 | 15.50 | 15.53 | 1.93 |
| | 55–64 | 20.49 | 18.05 | 1.43 | 20.19 | 17.79 | 1.41 | 20.66 | 18.20 | 1.45 |
| | 65–74 | 13.28 | 12.50 | 0.86 | 13.38 | 12.60 | 0.87 | 13.78 | 12.97 | 0.89 |
| | 75–84 | 4.77 | 4.83 | 0.42 | 4.58 | 4.64 | 0.40 | 4.94 | 5.00 | 0.43 |
| | 85 > = | 0.55 | 0.93 | 0.05 | 0.51 | 0.88 | 0.04 | 0.56 | 0.96 | 0.05 |
| | Total | 64.68 | 59.51 | 5.89 | 63.44 | 58.34 | 5.75 | 65.12 | 59.90 | 5.89 |
| **M + F** | - | 242.96 | 160.86 | 13.50 | 241.17 | 159.14 | 13.27 | 246.78 | 162.78 | 13.52 |
| **M%** | - | 73.3 | 63.0 | 56.3 | 73.6 | 63.3 | 56.6 | 73.6 | 63.2 | 56.5 |
| **F%** | - | 26.7 | 37.0 | 43.7 | 26.3 | 36.7 | 43.3 | 26.3 | 36.7 | 43.5 |

*Sat. Fat: Saturated fat

and 2019 were in the order of US$ 7.18 billion. The highest burden of CVD attributable were observed in younger individuals, which decrease with advancing age. Considering only salt as an inducer, an estimated 1.53 million expected productive years of life are lost due to premature death. Although attributable burden of CVD is higher among younger individuals, the highest costs are associated with men aged 45 to 74 years old for direct costs and 45 to 64 years old for indirect costs and expected years of productivity incurred because of permanent loss of productivity due to premature deaths.

Although the highest CVD costs attributable to the consumption of salt, saturated fat and trans fats is among the 45–74 age groups, the consumption of these nutrients is inversely proportional to the age. One of the explanations for our findings is that, due to the greater number of CVD cases occurring in these age groups, the risk attributable to the consumption of these nutrients is added to the risk related to the age group, suggesting that preventive measures at younger ages can benefit the entire population in their future.

Despite the continuous decline in crude and adjusted mortality rates for CVD (such as CHD and stroke), especially in the age group 35 to 44 years for both sexes [7], Brazil still has high mortality rates from these diseases and remains the leading cause of death in the country [40]. Unhealthy diet are one of the main causes, with an excess of critical nutrients such as salt, sugar, oils, and fats [27]. The increased consumption of UPFs is directly associated with the growth in consumption of these nutrients [13]. From 2000 to 2013, the traded volume of UPFs grew 43.7% worldwide and 48% in Latin America [15], so the intake of risk factor nutrients also increased [27].

Table 3. Indirect costs attributable to excess consumption of nutrients (salt, saturated fat and trans-fat) by sex and age group in US$.

| Year | Indirect Costs | Male (M) | | | | | M + F |
|------|----------------|----------|----------|----------|----------|----------|-------|
| | | 25–34 years | 35–44 years | 45–54 years | 55–65 years | Total | |
| 2017 | Salt Costs | 140.156.876 | 379.619.463 | 689.564.809 | 522.422.390 | 1.731.763.539 | 2.157.206.808 |
| | Sat. Fat* Costs | 105.289.015 | 273.682.097 | 459.250.077 | 260.911.322 | 1.099.132.511 | 1.472.231.054 |
| | Trans-Fat Costs | 22.843.454 | 35.492.147 | 45.496.694 | 14.125.903 | 117.958.197 | 164.782.112 |
| 2018 | Salt Costs | 133.183.858 | 370.817.683 | 700.298.668 | 523.168.842 | 1.727.469.050 | 2.142.931.279 |
| | Sat. Fat* Costs | 100.050.726 | 267.336.559 | 466.398.825 | 261.284.119 | 1.095.070.229 | 1.458.929.613 |
| | Trans Fat Costs | 21.706.957 | 34.669.233 | 46.204.902 | 14.146.086 | 116.727.178 | 162.267.613 |
| 2019 | Salt Costs | 138.312.820 | 383.595.663 | 685.977.191 | 532.698.544 | 1.740.584.217 | 2.153.450.921 |
| | Sat. Fat* Costs | 103.903.718 | 276.548.691 | 456.860.723 | 266.043.500 | 1.103.356.632 | 1.464.909.893 |
| | Trans Fat Costs | 22.542.900 | 35.863.898 | 45.259.987 | 14.403.762 | 118.070.547 | 163.303.520 |
| Year | Indirect Costs | Female (F) | | | | | M%—F% |
| | | 25–34 years | 35–44 years | 45–54 years | 55–65 years | Total | |
| 2017 | Salt Costs | 46.529.998 | 112.907.731 | 162.654.562 | 103.350.978 | 425.443.269 | 80.2% - 19.7% |
| | Sat. Fat* Costs | 34.019.473 | 85.140.776 | 162.909.522 | 91.028.771 | 373.098.543 | 74.6% - 25.3% |
| | Trans Fat Costs | 7.734.167 | 11.633.291 | 20.225.925 | 7.230.532 | 46.823.914 | 71.5% - 28.4% |
| 2018 | Salt Costs | 45.242.926 | 110.990.110 | 155.338.898 | 103.890.295 | 415.462.229 | 80.6% - 19.4% |
| | Sat. Fat* Costs | 33.078.457 | 83.694.748 | 155.582.391 | 91.503.787 | 363.859.383 | 75.1% - 24.9% |
| | Trans Fat Costs | 7.520.231 | 11.435.711 | 19.316.230 | 7.268.263 | 45.540.435 | 71.9% - 28.1% |
| 2019 | Salt Costs | 44.789.505 | 110.516.817 | 154.114.749 | 103.445.633 | 412.866.704 | 80.8% - 19.2% |
| | Sat. Fat* Costs | 32.746.946 | 83.337.851 | 154.356.324 | 91.112.140 | 361.553.261 | 75.3% - 24.7% |
| | Trans Fat Costs | 7.444.864 | 11.386.946 | 19.164.008 | 7.237.154 | 45.232.972 | 72.3% - 27.7% |

*Sat. Fat: Saturated fat

Table 4. Years of Life Lost (YLL) due to premature death by sex and age group.

| Sex | Age | 2017 | | | 2018 | | | 2019 | | |
|-----|-----|------|----------|-----------|------|----------|-----------|------|----------|-----------|
| | | Salt | Sat. Fat* | Trans fat | Salt | Sat. Fat* | Trans Fat | Salt | Sat. Fat* | Trans fat |
| Male | 25–34 | 43.368 | 32.579 | 7.068 | 41.131 | 30.899 | 6.704 | 42.431 | 31.875 | 6.916 |
| | 35–44 | 99.092 | 71.439 | 9.264 | 96.169 | 69.332 | 8.991 | 97.903 | 70.582 | 9.153 |
| | 45–54 | 148.116 | 98.645 | 9.772 | 148.503 | 98.903 | 9.798 | 145.658 | 97.008 | 9.610 |
| | 55–65 | 84.056 | 41.980 | 2.273 | 84.036 | 41.970 | 2.272 | 85.702 | 42.802 | 2.317 |
| | Total | 374.632 | 244.643 | 28.378 | 369.839 | 241.103 | 27.765 | 371.694 | 242.267 | 27.997 |
| | Age | 2017 | | | 2018 | | | 2019 | | |
| | | Salt | Sat. Fat* | Trans fat | Salt | Sat. Fat* | Trans Fat | Salt | Sat. Fat* | Trans fat |
| Female | 25–34 | 21.089 | 15.419 | 3.505 | 20.957 | 15.322 | 3.483 | 21.003 | 15.356 | 3.491 |
| | 35–44 | 44.341 | 33.436 | 4.569 | 43.678 | 32.936 | 4.500 | 42.536 | 32.076 | 4.383 |
| | 45–54 | 53.392 | 53.476 | 6.639 | 50.868 | 50.948 | 6.325 | 50.434 | 50.513 | 6.271 |
| | 55–65 | 24.318 | 21.418 | 1.701 | 24.335 | 21.434 | 1.702 | 24.383 | 21.476 | 1.706 |
| | Total | 143.140 | 123.750 | 16.415 | 139.837 | 120.640 | 16.012 | 138.357 | 119.421 | 15.851 |
| M + F | - | 517.772 | 368.393 | 44.793 | 509.677 | 361.743 | 43.777 | 510.051 | 361.688 | 43.848 |
| M% | - | 72.3 | 66.4 | 63.3 | 72.5 | 66.6 | 63.4 | 72.8 | 66.9 | 63.8 |
| F% | - | 27.7 | 33.6 | 36.7 | 27.5 | 33.4 | 36.6 | 27.2 | 33.1 | 36.2 |

*Sat. Fat: Saturated fat

Nilson et al. (2021) [24] estimated the economic effects and impact on health between 2013 and 2032 of the implementation of sodium reduction in processed foods in Brazil. They also reported that during this period, around 110,000 CVD male cases and 70,000 CVD female cases will be prevented. This estimate will result in a total savings of around US $ 220 million in medical costs for the Brazilian National Health System for the treatment of CHD and stroke. Moreover, corroborating our findings, a recent study carried out in Brazil showed that the costs attributable to excessive salt consumption were higher for males when compared to females, corresponding to 62% of the costs associated with hospitalizations and 53% of outpatient's costs for CVD, attributable to salt consumption [27].

According to Mozaffarian et al. (2006) [41], a 2% increase in trans fat consumption can increase the risk of CHD by up to 23%. Therefore, several countries have sought to regulate policies to limit the industrial use of trans fat [42] and to obligatorily include the content of this nutrient on food labels [43]. Marklund et al. (2020) [44] estimated that the ban on the use of trans fats would prevent 2,294 and 9,931 events of CHD in the first 10 years in Australia. In Argentina, where there was voluntary adherence to the reduction of trans fat in foods by the industry [45], according to Rubinstein et al. (2015) [46], given an annual incidence rate of almost 500 cases per 100,000 individuals over 34 years of age, the zero implementation of trans fats by the industry would prevent between 1.3% and 6.35% of cases of CHD, generating savings of, respectively, US$17 million and US$87 million.

Regarding the consumption of saturated fats, Dall et al. (2009) [47] estimated that if saturated fat intake reached 4g for adults with LDL-C > 100 mg/dL, the savings generated in CVD costs would be US$443.8 per case per year in the United States. In Germany, direct expenditure on these diseases, caused by excessive consumption of saturated fat, amounts to 2.9 billion euros a year [48]. The food replacement of this type of fat by polyunsaturated fats leads to a decrease in the risk of ischemic heart disease [49], therefore, being a possible option to minimize costs and losses in productivity generated, directly and/ or indirectly by illness.

In a study developed in Germany which relates the direct costs with non-communicable diseases [48], the highest expenses were attributed to the consumption of sugar (EUR 8.6 billion), followed by the consumption of salt (EUR 5.3 billion) and saturated fat (EUR 2.9 billion). The same pattern was observed in this study, in relation to salt (US$ 730.91 million) and saturated fat (US$ 482.78), considering that sugar was not analyzed.

In addition to the estimated years of life lost and costs associated with CVD (such as CHD and stroke), premature deaths from these diseases also cause several issues not addressed in this study. For instance, a recent study carried out in Brazil by Camarano (2020) [50], showed that premature loss of elderly victims due to COVID-19, in an extreme situation of this entire population dying, the household income monthly per capita would go from BRL 1,772.2 to BRL 529.2. This means a decrease of almost 75%, affecting 12.1 million people, of which 2.2 million are under 15 years old. Therefore, it can be inferred that the socioeconomic scenario of premature mortality from CVD is even more catastrophic, since the average age of death is 54 years old, with the leading ratio being from 55 to 65 years of age for both sexes. Thus, premature deaths from CVD should leave hundreds of thousands of younger people despondent, with greater vulnerability, due to their dependence on the income of older people. On the other hand, the earlier the death, the greater the individual indirect costs are due to the longer loss of potential productivity and yield of human capital.

In the American Continent, several countries have undergone food regulations. Frontal food labeling is being adopted in Chile, Peru, and Uruguay [51]. Brazil passed a bill for frontal labeling of foods in the year 2020 that should be effective as of October 2022, 24 months later, displaying a warning of the presence of excess added sugar, salt and fat, in addition to including in the table of nutritional composition of industrialized foods the amount of sugar added

to the product [52]. This measure is an important step to alert consumers about the nature of the products chosen on the market, making it possible to make healthier choices.

Regarding CVD prevention policies and programs in Brazil, every ten years the Ministry of Health prepares a plan of strategic actions to 'ht non-communicable diseases and conditions. The last document concerns the period of 2021 to 2030 and defends the increase of physical activity practice, consumption of healthy foods, reduction of smoking and alcohol consumption in the population [53]. To achieve the goal of promoting healthy eating, it includes the proposal to identify technical-scientific and political subsidies to support the elaboration of regulatory and fiscal measures to reduce the consumption of ultra-processed foods and encourage the consumption of in natura and minimally processed foods [53].

Studies that measure the economic impact attributable to salt, saturated fats, and trans fats, as is the case in the present study, are important to reinforce the importance of policies that enable the reduction intake of these nutrients and minimize their socioeconomic effects without loss of social well-being due to this restriction, between the sexes and the most affected age groups.

Brazil performed only the voluntary regulation of salt in processed foods [54]. One of the limitations of this positioning is not always reaching the defined goals, as well as not reaching all products available on the market. Furthermore, the absence of legal instruments to guarantee the achievement of targets and eventually impose sanctions in case of non-compliance is also an issue that should be taken into consideration [55]. In addition to these two matters, it is still important to consider that, according modeling study developed by Nilson et al. [56], the adoption of official regulatory limits with the lowest global targets could reduce salt consumption by 0.8g/day, compared to a reduction of 0.25g/day that is achieved with the current Brazilian voluntary targets, and could prevent about four times more deaths in ten years than the actual measure.

Furthermore, according to the revision of the Food Guideline for the Brazilian Population [57] there was considerable progress on the quality of food consumed, as well as an increase in the number of gyms at public parks. These are usually referred as Open Air Gyms, which were implemented by city halls all over the country [7], playing a vital role in the health of the population. However, there is a constant need for attention and confrontation of NCD, especially the CVD. This is because of their greater mortality compared to others, which may require specific measures that enable the improvement of the control of dietary risk factors, and others that improve collective health.

This scenario shows that in the case of Brazil, despite the great scientific advance in the recognition of risk factors associated with CVD, as in the case of this and other recent published works, and important official documents such as the aforementioned action plan or the new food guideline, there is no official restriction or taxation action for products with high amounts of salt, fat or sugar. In Mexico, sweetened beverages were taxed [58], and Argentina has already advanced observing regulatory measures to reduce sodium [59].

Countries such as the United Kingdom (UK) and Turkiye had a successful implementation of sodium reformulation policies [60], while the United States finds it difficult to include this type of policy, due to the strong rejection from the industry [61]. The UK showed a different scenario with its salt reduction policy, the implementation of this policy resulted in a 7% reduction of sodium in processed foods [62], while the industry continued to grow [24].

Finally, according to Ezzati et al. (2015) [63], the control of variable risk factors, such as diet, can contribute to a 50% reduction in mortality from CVD. In Brazil, Nilson et al. (2020) [23] estimated in 2017 that if the average daily consumption of sodium was reduced to 2g, 46,651 deaths from CVD would be avoided.

To our knowledge, no study has been previously carried out in Brazil, involving the loads attributable to the consumption of multiple nutrients in relation to cardiovascular diseases. This study brings scientific evidence that make strong arguments for the political decision to promote the consumption of healthy diets, reinforce the importance of food labeling and the taxation policies in relation to ultra-processed foods in the country.

However, there are some limitations of this study that should be highlighted. First, we only collected information's on public expenditure, we did not observe private health expenditures. We estimate the direct costs adding to the expenses recorded in the two official databases SIH/SUS and SIA/SUS. Although there is a greater participation of private spending in the acquisition of health services in Brazil, public spending represents approximately 40% of the total expenditure. We emphasize that observing only public spending is a limitation of our data. Second, a longitudinal assessment of the direct and indirect costs is essential to better understand the trends related to these expenses and losses.

Another aspect that should be mentioned is the estimation of indirect costs. There is a high probability that those values are being underestimated, for two main reasons: first, due to fact that the variables of the state of the country, marital status and education level remain constant overtime. Therefore, the earlier the death of the individual, the greater the underestimation of their loss of future earnings; and second, 2015 was the year which was used by the NHSS for the individual income, Brazil was experiencing an economic crisis that impacted employment and consequently, the income of the population, which can reflect on the results found in this study. Besides, the fact that other studies use different approaches to calculate the indirect costs makes it difficult to compare the results [12, 64, 65].

## Conclusions

Studies of this nature are important for Brazil's economy and the world society, being a necessary tool for the formulation of public policies that enable the minimization of involvement and effects by NCDs, especially CVD. Firstly, the study provides useful data that measures the impacts on public spending, and the social and productivity losses that negatively reflect on the economy. Additionally, it allows the comparison of pre-public and post-political intervention values for health systems and the socioeconomic response of these measures. This is possible via effective methods of measuring the impact of reducing the consumption of dietary risk factors, such as those analysed in this work.

In conclusion, the attributable fractions to consumption of salt are the ones that cause the most effects on CVD, followed by saturated fat and trans fat, with direct and indirect costs being higher for males.

## Author Contributions

**Conceptualization:** Jevuks Matheus de Araújo, Adélia da Costa Pereira de Arruda Neta, Flávia Emília Leite Lima Ferreira, Rafaela Lira Formiga Cavalcanti de Lima, Rodrigo Pinheiro de Toledo Vianna, José Moreira da Silva Neto, Patrícia Vasconcelos Leitão Moreira.

**Data curation:** Jevuks Matheus de Araújo, Rômulo Eufrosino de Alencar Rodrigues, Adélia da Costa Pereira de Arruda Neta, Patrícia Vasconcelos Leitão Moreira.

**Formal analysis:** Jevuks Matheus de Araújo, Rômulo Eufrosino de Alencar Rodrigues, Adélia da Costa Pereira de Arruda Neta, Patrícia Vasconcelos Leitão Moreira.

**Funding acquisition:** Patrícia Vasconcelos Leitão Moreira.

**Methodology:** Jevuks Matheus de Araújo, Adélia da Costa Pereira de Arruda Neta, Flávia Emília Leite Lima Ferreira, Rafaela Lira Formiga Cavalcanti de Lima, Rodrigo Pinheiro de Toledo Vianna, José Moreira da Silva Neto, Patrícia Vasconcelos Leitão Moreira.

**Project administration:** Patrícia Vasconcelos Leitão Moreira.

**Supervision:** Patrícia Vasconcelos Leitão Moreira.

**Writing – original draft:** Jevuks Matheus de Araújo, Rômulo Eufrosino de Alencar Rodrigues, Adélia da Costa Pereira de Arruda Neta, Flávia Emília Leite Lima Ferreira, Rafaela Lira Formiga Cavalcanti de Lima, Lucas Vasconcelos Leitão Moreira, José Moreira da Silva Neto, Patrícia Vasconcelos Leitão Moreira.

**Writing – review & editing:** Adélia da Costa Pereira de Arruda Neta, Lucas Vasconcelos Leitão Moreira, Patrícia Vasconcelos Leitão Moreira.

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
