## [Decision Letter · Decision Letter 0]

1 Aug 2022

PONE-D-21-41019The direct and indirect costs of cardiovascular diseases in BrazilPLOS ONE

Dear Dr. Moreira,

Thank you for submitting your manuscript to PLOS ONE. After careful consideration, we feel that it has merit but does not fully meet PLOS ONE’s publication criteria as it currently stands. Therefore, we suggest a major revision and invite you to submit a revised version of the manuscript that addresses the points raised during the review process.

We look forward to receiving your revised manuscript.

Kind regards,

Pankaj Bahuguna, Ph.D.

Guest Editor

PLOS ONE

Journal Requirements:

Additional Editor Comments:

Line 94-95, please provide brief description of 'National Health System Database', Ministry of Health Brazil in terms of what does it covers; since when this data on direct costs is available in dataset; primary source of data in dataset; is it in public domain? etc. This information will be helpful for non-Brazilian readers.I suggest to add brief description of methods used Ywata *et al*. (2008) for estimating indirect costs to make it reader friendly.At present, authors only give conclusions at the end of discussion. I suggest to add some clear recommendations in relation to public policy.

Reviewers' comments:

Reviewer's Responses to Questions

**Comments to the Author**

1. Is the manuscript technically sound, and do the data support the conclusions?

Reviewer #1: Yes

Reviewer #2: Yes

2. Has the statistical analysis been performed appropriately and rigorously? 

Reviewer #1: I Don't Know

Reviewer #2: I Don't Know

3. Have the authors made all data underlying the findings in their manuscript fully available?

Reviewer #1: No

Reviewer #2: Yes

4. Is the manuscript presented in an intelligible fashion and written in standard English?

Reviewer #1: No

Reviewer #2: No

5. Review Comments to the Author

Reviewer #1: Reviewer Comments to authors

General comments

Thank you for the opportunity to review this manuscript. Overall, this is an important area of research and imperative to know the costs involved in cardiovascular diseases that are cause of high morbidity and mortality globally. The paper details on the costs involved but lacks clarity on costing methods used and standard terminologies for direct costs, unit costs involved and how authors arrived at them. Focus is more outlined for indirect costs, however both are important in cardiovascular diseases. For the better clarity and understanding of the readers, the language editing is required at multiple places in the current manuscript.

Specific comments

Abstract

Line 26: It would be helpful to mention the study setting in the objective.

Methods: Which of the costing approach was used in the study and what was the perspective of the costing exercise? Please add few details in the methodology.

Results: Please rephrase the line 36 for better clarity.

Line 40: Please add appropriate unit with age group defined.

Introduction

Line 48: Please clarify whether it is mortality rate or a different indicator where authors refer to lethality rate. It is unclear.

Materials and methods

Line 89 is unclear. What is implied by databases have time limits? Did it have any effect on the main analysis and final outcomes? It would be good to clarify here than referring elsewhere.

Which costing approach and perspective was used for costing. Please provide details in the methodology section.

Line 112-113: How was this ideal consumption identified? Please provide supporting reference and the reference standard taken.

Line 120: The authors mention that food consumption was measured on non-consecutive days. Please provide details on choosing this rationale. Could this have introduced any bias and affected the analysis? Why were a sub sample chosen? It is unclear how many measurements and how many times in all at individual level were done to capture food consumption from the current manuscript.

Line 126: It is unclear on whom and how the adherence test was done.

Line 132: It would be good to mention the components of public expenses here for better understanding of the readers.

Line 138-139: Were there any individuals who had both both CHD and stroke? How were these cases if there handled in the cost analysis?

Results:

Line 185-186: Is this the study finding or finding from literature review. It is unclear. If the latter then could be shifted to discussion.

Discussion

In general authors have extensively reviewed about various aspects but this section does not clearly outline key findings and their implications and is quite confusing to read. Few suggestions that could be helpful to authors to strengthen this very important aspect of the paper for better clarity and understanding to readers are:

Introductory paragraph outlining what the study entailed, what was found, and why this is important in place of citing other paper on study design

Summarising key findings of current study

Importance of key findings in terms of what they tell us and implications of findings, in the context of what is already known in the literature and preferably in similar settings, and what is novel.

Study limitations are mentioned. However, the authors mentioned in methods about limited time-limit of databases. Were there any limitations in for the data availability or health information management systems? Were any limitations present in relation to estimation of direct costs?

Future direction/studies (optional)

Reviewer #2: Introduction general comment: Can you specifically outline the policy rationale for undertaking this analysis looking at this specific relationship between nutrients, CVDs and costs

It would be helpful for the reader to understand Brazilian sociodemographic characteristics specifically in terms of the population age characteristics to contextualize results better?

Line 55: Could you please specify what the respective age scenarios are that you are referring to?

Line 58: A brief overview of financing of Brazilian healthcare system along with these statements might be helpful

Line 66: Please consider clearly rephrasing statements when you are referring to author names. Also, it would be helpful to know what these authors have concluded in their respective studies.

Line 67: Grammatical error: ‘others’ should be replaced with ‘other’. Errors in structuring of sentences are quite common throughout the manuscript. It would help you to review these throughout the manuscript and to not use longer complex sentences.

Line 69: ‘the’ needs to be replaced by ‘a’

Line 76: Could you give more evidence and references to back this claim?

Line 79: What is the method generally used to know indirect costs?

Line 81: Premature death cases are only due to CVD? How has this been established?

Materials and Methods general comment: It would be helpful to outline the extent of indirect costs you are considering in the analysis with clear justification for doing so and providing a reason for excluding components of indirect costs that are not analysed in this analysis

Line 89: I am not sure if the limits of databases have been adequately highlighted later in the manuscript or in limitations?

Line 110: Could you please mention the currency exchange rate and its source as a reference?

Line 116: How has individual food consumption been estimated from household food consumption dataset?

Line 124: What is the multiple source method mentioned here?

Line 132: Given that public expenses for CHD and stroke from available data sources are used, and that private expenses do account for healthcare costs in Brazil, how does this justify the projected costs?

Line 148: Are costs only due to premature deaths accounted for? What about the costs due to morbidity/disability? Has this been accounted for in the analysis?

Line 165: It is unclear to me why discounting has been used only for indirect costs given in the corresponding formula?

Table 2: Any specific reason for as to why trans-fat consumption is relatively high in female population as compared to males?

Discussion general comment: The relevance of these findings needs to be better contextualised to relevant policies in Brazil. Please identify and discuss specific aspects of policies that can be addressed given the findings of this study.

Line 309: 110,000 and 70,000 cases of what?

6. PLOS authors have the option to publish the peer review history of their article (what does this mean?). If published, this will include your full peer review and any attached files.

Reviewer #1: No

Reviewer #2: No

---

## [Author Response · Author response to Decision Letter 0]

15 Sep 2022

PLOS ONE

Dear Pankaj Bahuguna, Ph.D.

We would like to thank you and the reviewers for your time to provide feedback on our paper. We have addressed all Journal requirements and the reviewers’ comments below. 

Journal Requirements

Comments to the Author: 

1. “Please ensure that your manuscript meets PLOS ONE's style requirements, including those for file naming. The PLOS ONE style templates can be found at 

https://journals.plos.org/plosone/s/file?id=wjVg/PLOSOne_formatting_sample_main_body.pdf and https://journals.plos.org/plosone/s/file?id=ba62/PLOSOne_formatting_sample_title_authors_affiliations.pdf.”

Response: Thank you so much for your directions. Now the manuscript meets the required template styles.

2. “Please include your full ethics statement in the ‘Methods’ section of your manuscript file. In your statement, please include the full name of the IRB or ethics committee who approved or waived your study, as well as whether or not you obtained informed written or verbal consent. If consent was waived for your study, please include this information in your statement as well.”

Response: Thank you for spotting this omission. Now in the ‘Methods’ section, the full name of the committee is included. In addition, the source of the information was now added, and that the committee also waived the requirement for informed consent in this study.

Lines 132-135:

“The research protocol was approved by The Ethics Committee of Federal University of Paraíba with consent number 3.843.739. As these are secondary data made available on public domain sites of the Brazilian Unified Health System, ethics committee waived the requirement for informed consent.”

3. “In your Data Availability statement, you have not specified where the minimal data set underlying the results described in your manuscript can be found. PLOS defines a study's minimal data set as the underlying data used to reach the conclusions drawn in the manuscript and any additional data required to replicate the reported study findings in their entirety. All PLOS journals require that the minimal data set be made fully available. For more information about our data policy, please see http://journals.plos.org/plosone/s/data-availability.

Important: If there are ethical or legal restrictions to sharing your data publicly, please explain these restrictions in detail. Please see our guidelines for more information on what we consider unacceptable restrictions to publicly sharing data: http://journals.plos.org/plosone/s/data-availability#loc-unacceptable-data-access-restrictions. Note that it is not acceptable for the authors to be the sole named individuals responsible for ensuring data access.”

Response: Thank you for pointing out this omission. The microdata used in the study are available in public domain and open access databases. These databases were inserted in the text and in the list of references with the respective URLs, as follows:

 Brazilian Institute of Geography and Statistics: (https://www.ibge.gov.br/estatisticas/sociais/saude/24786-pesquisa-de-orcamentos-familiares-2.html?=&t=microdados); (https://www.ibge.gov.br/estatisticas/sociais/populacao/9127-pesquisa-nacional-por-amostra-de-domicilios.html?=&t=microdados); and (https://www.ibge.gov.br/estatisticas/sociais/populacao/9126-tabuas-completas-de-mortalidade.html?=&t=downloads. 

 Informatics Department of the Brazilian Unified Health System (DATASUS) for the Outpatient Information System (SIA, specifically, SIA-PA [production subsystem]), Hospital Information System (SIH, specifically, SIA-RD [subsystem of accepted admissions]) and Mortality Information System (SIM, specifically, SIM-DO [death certificate subsystem], all on the same link for access to raw data (https://datasus.saude.gov.br/transferencia-de-arquivos/). 

Additional Editor Comments:

“Line 94-95, please provide brief description of 'National Health System Database', Ministry of Health Brazil in terms of what does it covers; since when this data on direct costs is available in dataset; primary source of data in dataset; is it in public domain? etc. This information will be helpful for non-Brazilian readers.” 

Response: Thank you for your suggestion. DATASUS is an open public domain secondary database that includes a wide range of systems and subsystems that contain time-varying information between them. Taking the Outpatient Information System (SIA) as an example, it is subdivided into 12 subsystems, the one used by us was the SIA-PA which deals with information on outpatient production and its values in monetary terms, with data availability 1994. The other two systems used, SIH and SIM, also follow this same model, with data available, respectively, from 1992 and 1979. These details are now contained in the manuscript.

In Lines 109-113, now it says: 

“DATASUS is an open public domain secondary database that includes a wide range of systems and subsystems that contain time-varying information between them, and a variety of information systems on Brazilians in relation to mortality, hospital care, outpatient care, basic health care, live births, physical structure of hospitals and clinics, professionals included in the SUS, notifications of contamination of epidemiological diseases, among others [26].” 

“I suggest to add brief description of methods used Ywata et al. (2008) for estimating indirect costs to make it reader friendly.” 

Response: Thank you for your suggestion. Now, a brief description of the methods in the manuscript was added.

Lines 187-196, now it reads: 

“Using the income data of individuals and crossing these data with information regarding age group, sex, schooling, and geographic location of the residence, it is possible to establish an estimate of the average income for subgroups of the population [28]. To obtain the income stream of these population subgroups we project the estimated income over time. Thus, to understand the loss of productivity, the reduction of income generated by the deaths caused by the CHD and stroke imputed to the subgroups of the dead population the estimated income stream for the same subgroups of the population that maintained their work activities. The estimates of average income and projection of future income follow the methodology described in Ywata et al [33]. To separate the causes, we weighed the income stream of each population subgroup by the PAF.”

“At present, authors only give conclusions at the end of discussion. I suggest to add some clear recommendations in relation to public policy.” 

Response: Thank you so much for highlighting this point. The entire discussion was reviewed by the authors and specific public policy points were added, as suggested. In Lines 293-423.

Reviewer #1

General comments to the Author: 

“Thank you for the opportunity to review this manuscript. Overall, this is an important area of research and imperative to know the costs involved in cardiovascular diseases that are cause of high morbidity and mortality globally. The paper details on the costs involved but lacks clarity on costing methods used and standard terminologies for direct costs, unit costs involved and how authors arrived at them. Focus is more outlined for indirect costs; however both are important in cardiovascular diseases. For the better clarity and understanding of the readers, the language editing is required at multiple places in the current manuscript.”

Response: Thank you so much for taking your time to contribute to the improvement of our manuscript. Your contributions were valuable in bringing greater clarity to the manuscript and in improving its overall quality and organization. The entire text has been revised in its language to facilitate the understanding and for better clarity of the readers.

Specific comments to the Author: 

Abstract

“Line 26: It would be helpful to mention the study setting in the objective. 

Methods: Which of the costing approach was used in the study and what was the perspective of the costing exercise? Please add few details in the methodology. 

Results: Please rephrase the line 36 for better clarity.

Line 40: Please add appropriate unit with age group defined.”

Response: Thank you for the suggestions in relation to the abstract.

Objective:

In Lines 21-23, now it says: 

“Objective: To evaluate the direct and indirect costs of cardiovascular diseases (such as coronary heart disease and stroke) by sex and age group, attributed to the excessive consumption of salt, saturated fat and trans fats in Brazil.”

Methods:

In Lines 27-32, now it says:

“Methods: The calculation of direct costs was made from the accounting sum of hospital costs with hospitalizations and outpatient care found in the Hospital Information System and Outpatient Information System. Regarding the indirect costs, they were measured by the loss of human capital, given the premature death, via a non-parametric method of pairing by observational explanatory characteristics for income between the living and the dead. To define the attributable costs, they were multiplied by the PAF.”

Results:

In Lines 33-38, now it says:

“Results: Higher burdens attributable to the consumption of salt, saturated fat and trans fat were observed in younger individuals, which progressively decreased with advancing age, but still generated economic costs in the order of US$ 7.18 billion, in addition to 1.53 million productive years of life lost to premature death, if considering salt as an inducer. Although attributable burdens are higher among younger individuals, the highest costs are associated with men aged 45 to 74 years old for direct costs and 45 to 64 years old for indirect costs.”

Introduction

“Line 48: Please clarify whether it is mortality rate or a different indicator where authors refer to lethality rate. It is unclear.”

Response: Thank you. Now, in Lines 44-47, it says: 

“Cardiovascular diseases (CVDs) are one of the great challenges of world health, since they are the main cause of premature deaths and compromised productivity, reaching a mortality rate in 2019 of around 32% in the total population and 38% in individuals under the age of 70 [1–3].”

Materials and methods

“Line 89 is unclear. What is implied by databases have time limits? Did it have any effect on the main analysis and final outcomes? It would be good to clarify here than referring elsewhere.

Which costing approach and perspective was used for costing. Please provide details in the methodology section.” 

Response: Thank you for your comment. Our apologies for not being clear about the sentence “some databases have time limits that will be highlighted.” We rewrote the sentence and explained the issue of databases better by talking about the limitations in the discussion section. 

Now, in Lines 99-100, it says:

“The analysis of CVD cost information for Brazil was carried out from 2017 to 2019, using several databases that will be highlighted.”

“Line 112-113: How was this ideal consumption identified? Please provide supporting reference and the reference standard taken.” 

Response: Thank you for your comment. We added the ideal consumption of each evaluated nutrient, as well as their respective references.

We added the statement:

In Lines 144-146:

“In this study, the TMREL for salt was 5g/day, for saturated fat was 10% of the total daily energy value and for trans-fat was 1% of the total daily energy value [8].”

“Line 120: The authors mention that food consumption was measured on non-consecutive days. Please provide details on choosing this rationale. Could this have introduced any bias and affected the analysis? Why were a sub sample chosen? It is unclear how many measurements and how many times in all at individual level were done to capture food consumption from the current manuscript.” 

Response: Thanks for your comment. The Personal Food Consumption Block, HBS 7, was the instrument used to record information on food intake. This instrument was developed with the participation of specialists from all over the country, based on a partnership established between the IBGE and the Ministry of Health. To carry out the food record, the individuals were instructed to record and report in detail the names of the foods consumed, the type of preparation, the measure used, the amount consumed, the time and whether the consumption of the food occurred at home or outside the home. In cases where there was an impediment for the resident to complete the registration, it could be done with the help of another resident or a close person.

Data referring to the personal food consumption module were collected for all residents aged 10 years and over from 20,112 selected households, which corresponded to a subsample of 34.7% of the 57,920 households investigated in the HBS 2017-2018. In this way, information was obtained about the individual food consumption of 34,003 residents. Households that participated in the subsample were randomly selected from among those households that were selected for the original HBS sample.

Although the food record provides for the completion of one day of food consumption, it can be applied in two or more days. A single food record per individual may be sufficient when the objective of the study is to estimate the average consumption of food and/or nutrients for a population group. However, most studies that assess food consumption aim to assess the distribution of individual consumption or investigate the proportion of individuals who inappropriately consume a group of foods or a particular nutrient or even analyze the association of dietary factors with the outcome of health. In all these cases, the interest is to assess the usual food consumption. Thus, a single day of food records is not capable of estimating an individual's eating habits. For this reason, replication of the method is important and necessary. In addition, it is believed that there is a dependence on food consumption on consecutive days, that is, the diet of a given day can influence the dietary consumption of the following day. Therefore, it is suggested to use consecutive days and preferably covering weekdays and weekends (Willett, 2013).

Statistical tools, such as the MSM, have been developed and increasingly improved to correct for intrapersonal variability and estimate habitual consumption from a limited number of food record days. For these tools to be used, it is necessary that the food record be applied in at least two days so that the intra-individual variability is estimated. Some statistical methods allow estimating distributions of habitual food consumption when two days of dietary assessment are obtained in 40 to 60% of the total sample, that is, it is not necessary to obtain two days of food records from all individuals investigated (Verly Jr et al., 2012).

References:

 Verly JR E, Castro MA, Firsberg RM, Marchioni DM. Precision of usual food intake estimates according to the percentage of individuals wiht a second dietary measurement. J Acad Nutr Diet. 2012; 112(7):1015-20.

 Willett WC. Nature of variation in diet. In: Willett WC. Nutritional epidemiology. 2. ed. New York: Oxford University Press; 2013. p. 34-48.

In Lines 155-160, it now says:

“Food consumption was measured using food records, applied on non-consecutive days, in which individuals were instructed to record and report in detail the names of the foods consumed, the type of preparation, the measure used, the amount consumed, the time and whether the consumption of the food occurred at home or outside the home, with their servings sizes converted from standard units or household measures, to grams, using a common reference table to HBS [14].”

“Line 126: It is unclear on whom and how the adherence test was done.” 

Response: One way of trying to verify whether or not a distribution fits well to the sample data is by comparing the sample frequencies with the theoretical frequencies expected by the probabilistic model, that is considered valid to describe the observed data. There are hypothesis tests, called goodness-of-fit tests, which serve to test more general hypotheses about the distribution of data. This means that they assess whether or not the distance from the distribution of observed data is significant in relation to a distribution of reference. In the case of the present study, adherence tests were performed in the R software to verify the distribution of food consumption variables (salt, saturated fat and trans fats).

We changed the text in the manuscript:

Now, it says in Lines 164-167:

“To verify the distribution that best fits the sample data, the adherence test was performed, which confirmed that the salt consumption data showed a log-normal distribution, while the saturated and trans fat data showed a gamma distribution. Both distributions were used in the PAF calculation.”

“Line 132: It would be good to mention the components of public expenses here for better understanding of the readers.” 

Response: Thank you for your suggestion. The text has been replaced by an explanation of the approaches commonly used in indirect cost analysis. 

Now, in Lines 86-89, it says:

“The main approaches in indirect cost analysis are human capital that is associated with lost productivity and friction costs that estimates the costs of worker replacement. We used the human capital approach and estimated the productivity losses based on wages over the working life of the worker.” 

“Line 138-139: Were there any individuals who had both CHD and stroke? How were these cases if there handled in the cost analysis?” 

Response: Thank you for your question. In the SUS database, there is no information on the individuals themselves, but on the count of specific morbidities and mortality by sex, age, and region. Therefore, it is not possible to know if the same individual is present in the CHD and Stroke counts simultaneously.

Results: 

“Line 185-186: Is this the study finding or finding from literature review. It is unclear. If the latter then could be shifted to discussion.” 

Response: Thanks for pointing this out. We removed the part of the text that confuses the interpretation, since we are talking about our results, which are presented in Table 1.

Now, in Lines 241-242, it says:

“The attributable fractions to salt are the ones that cause the most effects on CVDs, followed by saturated fat and trans fats.”

Discussion 

“In general authors have extensively reviewed about various aspects but this section does not clearly outline key findings and their implications and is quite confusing to read. Few suggestions that could be helpful to authors to strengthen this very important aspect of the paper for better clarity and understanding to readers are:

 Introductory paragraph outlining what the study entailed, what was found, and why this is important in place of citing other paper on study design

 Summarizing key findings of current study

 Importance of key findings in terms of what they tell us and implications of findings, in the context of what is already known in the literature and preferably in similar settings, and what is novel.

 Study limitations are mentioned. However, the authors mentioned in methods about limited time-limit of databases. Were there any limitations in for the data availability or health information management systems? Were any limitations present in relation to estimation of direct costs?

 Future direction/studies (optional)”

Response: Thanks for the comments. We have improved our discussion by considering the various points suggested above. We even added details about limitations related to databases. The new thread can be verified between the Lines 293-423.

Reviewer #2

Comments to the Author: 

“Introduction general comment: Can you specifically outline the policy rationale for undertaking this analysis looking at this specific relationship between nutrients, CVDs and costs

It would be helpful for the reader to understand Brazilian sociodemographic characteristics specifically in terms of the population age characteristics to contextualize results better?”

Response: Thanks for your comments. The introduction has been rewritten to take your suggestions into account. 

“Line 55: Could you please specify what the respective age scenarios are that you are referring to?” 

Response: Thank you for spotting this omission, we added the age scenario. 

Now, in the Lines 53-57, it says:

“Specifically in Brazil, between 2000 and 2018, crude mortality rates from CVD have been decreasing in adults over 25 years of age, of both sexes, except in men over 85 years of age [7]. Even in this scenario, CVDs were responsible for 28% of all deaths that occurred between 2010 and 2015, and 38% of this number occurred in the productive age group (18 to 65 years) [8].”

“Line 58: A brief overview of financing of Brazilian healthcare system along with these statements might be helpful”

Response: Thank you for your suggestion. 

Now, in Lines 58-65, it says:

“In addition to the irreversible social losses in the family environment, CVDs have a considerable weight in public and private financial costs (direct costs) due to hospitalizations, monitoring, treatment, and others, and in the loss of productivity (indirect costs). Health financing in Brazil comes from public and private sources. The model covers the Brazilian National Health System (SUS - Sistema Único de Saúde), supported by taxes and contributions collected at the federal, state and municipal levels, and the Complementary Health System, with resources from companies and individuals, with 71% of the Brazilian population using this system [9].” 

“Line 66: Please consider clearly rephrasing statements when you are referring to author names. Also, it would be helpful to know what these authors have concluded in their respective studies.”

Response: Thanks for raising these points. The text was rewritten, and the conclusions of the study were added.

“Line 67: Grammatical error: ‘others’ should be replaced with ‘other’. Errors in structuring of sentences are quite common throughout the manuscript. It would help you to review these throughout the manuscript and to not use longer complex sentences.”

Response: Thanks for raising these points. The manuscript has been extensively revised for grammatical issues.

“Line 69: ‘the’ needs to be replaced by ‘a’”

Response: Thanks for raising these points. The manuscript has been extensively revised for grammatical issues.

“Line 76: Could you give more evidence and references to back this claim?” 

Response: Thank you for your suggestion. We added some evidence and references from the literature. 

Now, in Line 71-77, it says:

“Nowadays, Brazilians have been going through gradual changes in eating patterns, evidently perceived on the excessive consumption of nutrients linked to the causes of Non communicable diseases (NCDs), such as sodium, sugars, and fats [13]. An increase in the consumption of ultra-processed foods (UPFs) by the Brazilian population has been observed [13,14]. These UPFs have high levels of the above mentioned nutrients, with scientific evidence of their relation to obesity [15,16], diabetes mellitus [17,18], hypertension [19] and cardiovascular diseases [20–22].”

References added below:

13. Louzada ML da C, Martins APB, Canella DS, Baraldi LG, Levy RB, Claro RM, et al. Ultra-processed foods and the nutritional dietary profile in Brazil. Rev Saúde Pública. 2015;49. doi:10.1590/S0034-8910.2015049006132

14. Brazilian Institute of Geography and Statistics. IBGE. Pesquisa de Orçamentos Familiares 2017-2018. Análise do Consumo Alimentar Pessoal no Brasil. IBGE; 2020. 

15. Pan American Health Organization. Ultra-processed food and drink products in Latin America: trends, impact on obesity, policy implications. Paho Washington (DC); 2015. 

16. Askari M, Heshmati J, Shahinfar H, Tripathi N, Daneshzad E. Ultra-processed food and the risk of overweight and obesity: a systematic review and meta-analysis of observational studies. Int J Obes 2005. 2020;44: 2080–2091. doi:10.1038/s41366-020-00650-z

17. Levy RB, Rauber F, Chang K, Louzada ML da C, Monteiro CA, Millett C, et al. Ultra-processed food consumption and type 2 diabetes incidence: A prospective cohort study. Clin Nutr Edinb Scotl. 2020. doi:10.1016/j.clnu.2020.12.018

18. Srour B, Fezeu LK, Kesse-Guyot E, Allès B, Debras C, Druesne-Pecollo N, et al. Ultraprocessed Food Consumption and Risk of Type 2 Diabetes Among Participants of the NutriNet-Santé Prospective Cohort. JAMA Intern Med. 2020;180: 283–291. doi:10.1001/jamainternmed.2019.5942

19. He FJ, Tan M, Ma Y, MacGregor GA. Salt Reduction to Prevent Hypertension and Cardiovascular Disease: JACC State-of-the-Art Review. J Am Coll Cardiol. 2020;75: 632–647. doi:10.1016/j.jacc.2019.11.055

20. Bonaccio M, Di Castelnuovo A, Costanzo S, De Curtis A, Persichillo M, Sofi F, et al. Ultra-processed food consumption is associated with increased risk of all-cause and cardiovascular mortality in the Moli-sani Study. Am J Clin Nutr. 2021;113: 446–455. doi:10.1093/ajcn/nqaa299

21. Pagliai G, Dinu M, Madarena MP, Bonaccio M, Iacoviello L, Sofi F. Consumption of ultra-processed foods and health status: a systematic review and meta-analysis. Br J Nutr. 2021;125: 308–318. doi:10.1017/S0007114520002688

22. Srour B, Fezeu LK, Kesse-Guyot E, Allès B, Méjean C, Andrianasolo RM, et al. Ultra-processed food intake and risk of cardiovascular disease: prospective cohort study (NutriNet-Santé). BMJ. 2019;365: l1451. doi:10.1136/bmj.l1451

“Line 79: What is the method generally used to know indirect costs?” 

Response: Thank you for your question. The text was replaced by an explanation of the approaches commonly used in indirect cost analysis. 

In Lines 86-89, it now says:

“The main approaches in indirect cost analysis are human capital that is associated with lost productivity and friction costs that estimates the costs of worker replacement. We used the human capital approach and estimated the productivity losses based on wages over the working life of the worker.”

“Line 81: Premature death cases are only due to CVD? How has this been established?” 

Response: Thank you. Deaths are filtered for CVD only. The premature deaths considered are those that occur under the age of 65 years, given that individuals have a share attributable to death from excessive consumption of salt, trans fat and saturated fat. In this way, the PAFs of these nutrients work as a weight that is assorts between 0 and 1, where 0 indicates no cost associated with these nutrients and 1 the total cost attributable to them. For clarity, the passage in question has been replaced.

In Lines 90-92, it now says:

“Observational variables are used that characterize the income of living individuals, correlating with the observational characteristics of individuals who had deaths from CVD under the age of 65 years.”

Materials and Methods general comment: “It would be helpful to outline the extent of indirect costs you are considering in the analysis with clear justification for doing so and providing a reason for excluding components of indirect costs that are not analysed in this analysis.” 

Response: Thank you for your suggestion. The choice to use the loss of permanent productive capacity given by the mortality came from three conditions: 1) Firstly, because in the indirect costs the individual's productivity resumption is irreversible; 2) Secondly, this way of estimating costs allows more realistic results without introducing possible bias. 3) Thirdly, because it is a very laborious task that takes a considerable amount of time, since the data used are not processed directly by the DATASUS system and, in addition, our approach includes age groups that are not available in it. Therefore, it was necessary to extract raw data and carry out all the processing by the statistical program R, in order to make the connection between the living and the dead, individual by individual. Then, we were able to separate the age groups present in the study and the sum of indirect costs. In this way, we made it impossible to add other indirect costs that could possibly bring bias to the results.

Three other indirect cost approaches were considered, but their limitations led us not to choose them:

- Approach 1: it would be developed using data from the Hospital Information System (SIH), which provides information on the number of days where individuals remained hospitalized, in other words, being unable to work and be productive, generating indirect costs. This approach would follow in a similar way to that found in the study, but without using the probability of death and without a discount rate, as the values would be in the present. However, the SIH does not provide valuable information for determining income, such as education and marital status of individuals (especially schooling, which is a recognized variable in the literature as a proxy for human capital and, consequently, for determining income), which could generate inaccurate information on the income of hospitalized individuals.

- Approach 2: simpler than the others, it would be the identification of permanently disabled and retired living individuals that would generate a direct social security cost, but indirectly due to premature productivity loss. This approach was disregarded because in most cases the amounts paid by social security are the minimum stipulated by law, so it is an exogenously determined income that would bias the results because it is controlled by the government and not by factors inherent to the individual.

- Approach 3: the last approach considered was the inverse of the one present in the study, which would be the attempt to estimate the marginal propensity of individuals to pay to live an additional year given premature death from excessive consumption of salt, trans fat and fat saturated, which is credited in the literature as the “value of life”. This line is used especially in studies on violence, but it has a strong tendency to overestimate indirect costs, which discouraged us, as there are data limitations for this estimation.

In Lines 233-238, it now says:

“This methodological approach to the estimation of indirect costs was considered due to the costs associated with permanent loss of productivity given the premature death from CVD, caused by excessive consumption of nutrients, being, in hypothesis, the highest within this cost modality [33]. In addition, the estimates are more realistic, as real observational variables are used as determinants of the income of deceased individuals, reducing possible bias in the results [33].”

“Line 89: I am not sure if the limits of databases have been adequately highlighted later in the manuscript or in limitations?” 

Response: Thank you for pointing out this issue. Now at the end of discussion section, it was added some limitations that occurred in this study.

In Lines 408-423, it now says:

“However, there are some limitations of this study that should be highlighted. First, we only collected information’s on public expenditure, we did not observe private health expenditures. We estimate the direct costs adding to the expenses recorded in the two official databases SIH/SUS and SIA/SUS. Although there is a greater participation of private spending in the acquisition of health services in Brazil, public spending represents approximately 40% of the total expenditure. We emphasize that observing only public spending is a limitation of our data. Second, a longitudinal assessment of the direct and indirect costs is essential to better understand the trends related to these expenses and losses. 

Another aspect that should be mentioned is the estimation of indirect costs. There is a high probability that those values are being underestimated, for two main reasons: first, due to fact that the variables of the state of the country, marital status and education level remain constant overtime. Therefore, the earlier the death of the individual, the greater the underestimation of their loss of future earnings; and second, 2015 was the year which was used by the NHSS for the individual income, Brazil was experiencing an economic crisis that impacted employment and consequently, the income of the population, which can reflect on the results found in this study.” 

“Line 110: Could you please mention the currency exchange rate and its source as a reference?” 

Response: Thank you for spotting this omission. The conversion rate used was 3.94 R$/U$S (0.25345 U$S/R$) which corresponds to the average of the 12 months of the year 2019. These data are available at the Institute of Applied Economic Research (IPEA) < http://www.ipeadata.gov.br/Default.aspx > from Brazil under the heading to access the data link “Exchange rate - R$ / US$ - commercial - purchase - average” by monthly data. The exchange rate information as well as its respective reference are now present in the manuscript.

In Lines 127-131, it now says:

“In addition, all monetary values were corrected using the Index National Price on Expanded Consumers (Índice Nacional de Preços ao Consumidor Amplo, IPCA/IBGE) in Brazilian currency (R$ - reais) for 2019 and, subsequently, converted to US dollars ($) at the average exchange rate for 2019, corresponding to 3.944 R$/U$S (0.25345 U$S/R$), made available by the Institute of Applied Economic Research (IPEA) [30].”

“Line 116: How has individual food consumption been estimated from household food consumption dataset?” 

Response: Thank you for your comment. The Personal Food Consumption Block, HBS 7, was the tool used to record the information on food intake. This tool was developed with the participation of specialists from all over the country, based on a partnership established between the IBGE and the Ministry of Health. To carry out the food record, the individuals were instructed to record and to report in detail, the following: the names of the foods consumed, the type of preparation, the measure used, the amount consumed, the time and whether the consumption of the food occurred at home or outside. In cases where there was an impediment for the resident to complete the registration, it could be done with the help of another resident or a close person.

Data referring to the personal food consumption module were collected from 20,112 selected households, for all residents with 10 years of age and over, which corresponded to a subsample of 34.7% of the 57,920 households investigated in the HBS 2017-2018. In this way, information was obtained about the individual food consumption of 34,003 residents. Households that participated in the subsample were randomly selected from among those households that were selected for the original HBS sample.

Although the food record provides for the completion of one day of food consumption, it can be applied in two or more days. A single food record per individual may be sufficient when the objective of the study is to estimate the average consumption of food and/or nutrients for a population group. However, most studies that assess food consumption aim to assess the distribution of individual consumption, or target to investigate the proportion of individuals who inappropriately consume a group of foods or a particular nutrient, or even aim to analyze the association of dietary factors with the outcome of health. In all these cases, the interest is to assess the usual food consumption. Thus, a single day of food records is not capable of estimating an individual's eating habits. For this reason, replication of the method is important and necessary. In addition, it is believed that there is a dependence on food consumption on consecutive days, that is, the diet of a given day can influence the dietary consumption of the following day. Therefore, it is suggested to use consecutive days and preferably covering weekdays and weekends (Willett, 2013).

Reference:

Willett WC. Nature of variation in diet. In: Willett WC. Nutritional epidemiology. 2. ed. New York: Oxford University Press; 2013. p. 34-48.

“Line 124: What is the multiple source method mentioned here?” 

Response: Thank you. Statistical tools, such as the Multiple Source Method (MSM), have been developed and increasingly improved, in order to correct for intrapersonal variability and estimate the habitual consumption from a limited number of food record days. Moreover, for these tools to be used, it is necessary that the food record be applied in at least two days so that the intra-individual variability is estimated. Some statistical methods allow estimating distributions of habitual food consumption when two days of dietary assessment are obtained in 40 to 60% of the total sample, that is, it is not necessary to obtain two days of food records from all individuals investigated (Verly Jr et al., 2012). We added the MSM´s reference in the manuscript. 

References:

 Verly JR E, Castro MA, Firsberg RM, Marchioni DM. Precision of usual food intake estimates according to the percentage of individuals with a second dietary measurement. J Acad Nutr Diet. 2012; 112(7):1015-20.

 Harttig, U et al. The MSM program: web-based statistics package for estimating usual dietary intake using the Multiple Source Method. European Journal of Clinical Nutrition, v.65, p. S87–S91, 2011.

The reference was added in the manuscript:

In Lines 161-163, it says:

“Habitual nutrient intake was estimated using the Multiple Source Method (MSM). This last method is suitable to estimate the usual individual intake for repeated measurements and a defined period [32].”

“Line 132: Given that public expenses for CHD and stroke from available data sources are used, and that private expenses do account for healthcare costs in Brazil, how does this justify the projected costs?” 

Response: Thank you. In this study, we collected only information’s on public expenditure, we did not observe private health expenditures. We estimate the direct costs adding to the expenses recorded in the two official bases SIH/SUS and SIA/SUS. Although there is a greater participation of private spending in the acquisition of health services in Brazil, public spending represents approximately 40% of the total expenditure. We emphasized that observing only public spending is a limitation of our data. 

“Line 148: Are costs only due to premature deaths accounted for? What about the costs due to morbidity/disability? Has this been accounted for in the analysis?” 

Response: Thank you for your comment. Yes, only premature death costs are counted. Unfortunately, there is no follow-up data over time for individuals with morbidity, so there is no way to identify how long the individual has had the morbidity or if he or she has ceased to have a state of morbidity. 

Furthermore, in addition to the need of data for the calculations, there would possibly be a need for an individual identification variable. For instance, the Individual Taxpayer Registry (CPF), is a unique number assigned to each Brazilian citizen. This would be necessary to avoid double counting of indirect costs. This is not the case for deaths, as each row in the raw database will always represent a death of a person and its observational characteristics. The only exception would be the time which the individual spent hospitalized and, consequently, absent from work, regardless of whether he had to be hospitalized “n” times, with the limitations mentioned above regarding the absence of important variables. In this case, it was only possible to calculate the direct costs that the morbidities generate for the SUS via hospitalizations and outpatient care. Regarding disability, there is the possibility of the individual retiring prematurely. In this condition, there would be data, however, the value would not correspond to the loss of productivity of the individual, but how much he contributed to the social security, which in most cases, results in retirement with a minimum wage stipulated by law. Thus, in both possible cases, it would produce biased results.

“Line 165: It is unclear to me why discounting has been used only for indirect costs given in the corresponding formula?” 

Response: Thank you for your comment. Our calculations for indirect costs were related to the permanent loss of productivity resulting from premature death. Calculations start at the current time extending into the future, under the assumption that the individual who dies in year t is alive in year t+1, t+2, until reaching the maximum productive age (considered to be 65 years). Thus, it is necessary to bring the future values t+1, t+2, until completing the entire productive age, to the present value. In this case, the values are calculated for the present values of 2017, 2018 and 2019. As the direct costs are current values for each year considered in the study, there is no need to use the discount rate. Furthermore, the discount rate used in the calculation of indirect costs was the same as in the study by Ywata et al. (2008).

Reference:

Ywata AXC et al. Custos das mortes por causas externas no Brasil. Rev Bras Biom. 2008;26: 23–47.

In Lines 214-218, now it says:

“To bring the future income stream to the present we apply the net present value (NPV) allows you to measure the value of money in time:

TICt,i=∑_(X=Di)^T▒1/〖(1+d)〗^((x-Di)) *Pr⁡(Fi>x│Fi≥Di)*Wi

In the occasion that (d) is the annual discount rate of 3%, (Wi) is the expected annual income of the individual (i) present in the SIM.” 

“Table 2: Any specific reason for as to why trans-fat consumption is relatively high in female population as compared to males?” 

Response: Thank you for your question. According to Household Budget Survey (POF 217/2018), which was the database for the present study, small differences were observed regarding the participation of the four food groups in the diet of men and women. The consumption of in natura or minimally processed foods was slightly higher in men than in women (54.1% and 52.8% of total calories, respectively) as well as consumption of processed foods (11.8% and 10.8%, respectively). On the other hand, the consumption of processed culinary ingredients was higher among women (16.2% against 15.0% among men) and the consumption of ultra-processed foods (20.3% against 19.1% among men). Therefore, one of the plausible explanations for the high consumption of trans fat among women is due to the consumption of ultra-processed foods, since these are the main sources of trans fat.

Discussion general comment: “The relevance of these findings needs to be better contextualised to relevant policies in Brazil. Please identify and discuss specific aspects of policies that can be addressed given the findings of this study.” 

Response: Thank you very much for your considerations. The discussion was reviewed by the authors and relevant policy aspects in Brazil were mentioned. Now it reads in Lines 293-423.

Line 309: 110,000 and 70,000 cases of what? 

Response: Thank you for spotting this omission. 

In Lines 310-314, it now says: 

“Nilson et al. (2021) [36] estimated the economic effects and impact on health between 2013 and 2032 of the implementation of sodium reduction in processed foods in Brazil. They also reported that during this period, around 110,000 CVD male cases and 70,000 CVD female cases will be prevented. This estimate will result in a total savings of around US $ 220 million in medical costs for the Brazilian National Health System for the treatment of CHD and stroke.”

Kind regards,

The author (On behalf of the co-authors)

---

## [Decision Letter · Decision Letter 1]

21 Oct 2022

PONE-D-21-41019R1

The direct and indirect costs of cardiovascular diseases in Brazil

PLOS ONE

Dear Dr. Moreira,

Thank you for submitting the revised manuscript to PLOS ONE. The revised manuscript has improved significantly but after careful consideration, we feel there is still scope of further improvement. Therefore, we invite you to submit a revised version of the manuscript that addresses the points raised during the review process.

We look forward to receiving your revised manuscript.

Kind regards,

Pankaj Bahuguna, Ph.D.

Guest Editor

PLOS ONE

Journal Requirements:

Reviewers' comments:

Reviewer's Responses to Questions

**Comments to the Author**

1. If the authors have adequately addressed your comments raised in a previous round of review and you feel that this manuscript is now acceptable for publication, you may indicate that here to bypass the “Comments to the Author” section, enter your conflict of interest statement in the “Confidential to Editor” section, and submit your "Accept" recommendation.

Reviewer #1: (No Response)

Reviewer #2: (No Response)

2. Is the manuscript technically sound, and do the data support the conclusions?

Reviewer #1: Partly

Reviewer #2: Yes

3. Has the statistical analysis been performed appropriately and rigorously? 

Reviewer #1: Yes

Reviewer #2: I Don't Know

4. Have the authors made all data underlying the findings in their manuscript fully available?

Reviewer #1: Yes

Reviewer #2: Yes

5. Is the manuscript presented in an intelligible fashion and written in standard English?

Reviewer #1: Yes

Reviewer #2: Yes

6. Review Comments to the Author

Reviewer #1: Thank you for revising the manuscript and improving clarity for better understanding of the readers. However, the paper still needs language editing at few places and use of shorter sentences.

Abstract

Line 24-25: It could be useful to know whether the cost mentioned here was overall cost/health system cost/societal cost and specifically for treatment?

Line 27-28: Were these direct medical costs only? Please specify if so.

Line 29-30 can be rephrased to shorter sentences for better clarity.

Results: Line 33: Higher burdens of what attributable to?

Conclusion Line 39: “Studies of this nature….” Is very broad scope and could be rephrased to narrower scope for clarity. Also authors may use shorter sentences in conclusion and should restrict to findings from the study than generic statements.

Introduction

Line 44: This line may be rephrased to “Cardiovascular diseases (CVD) are one of the key challenges in health globally, since……………”

Line 78: Please cite supporting references besides “few studies…….” Or may rephrase.

Line 85-86 looks repetitive in lines 94-96 and may be better placed there.

Lines 87-89 seems better suited to methodology section.

Line 99 could be rephrased to “Several databases were referred to collect and analyse CVD cost related information between 2017 and 2019 for Brazil.”

Line 114: There seems to be incomplete word in here. Whether its three sources or three databases, it is currently unclear to understand “three bases”.

Discussion

Line 293: The sentence could be rephased to “Findings from or study showed………….”

Reviewer #2: Thank you for considering suggestions and revising the manuscript. The manuscript content is structured better and reads more logically now. The authors have addressed grammatical mistakes to a large extent. Policy relevance of this work is highlighted better than the previous version. Following are some of my suggestions/clarifications for the revised manuscript (Line numbers refer to the latest unmarked version):

Consider further grammatical checks: For example, replace burdens with burden (multiple places, replace servings with serving – Line 158, typos on Line 214-215, Replace manuscript with ‘study’ – Line 293, typo error on Line 295, typo on Line 375, consider replacing ‘inadequate diet’ with maybe unhealthy diet if you thing that is more reflective on Line 305. You may consider further similar grammatical checks in the manuscript.

The revised introduction section makes a better read. However, on line 87, you go on to directly explain methods of indirect cost analysis. I would suggest you consider moving this to the methods section. The introduction section could give a brief overview of what all analysis (PAF, direct cost, indirect costs) is being aimed. Similarly, I found it difficult to establish relevance of Line 69-70 in context to the previous statement. What do you mean by recurrent costs here? My understanding is recurrent costs would be those attributed to hospitalisations and consultations (22% of costs)

For lines 137 to 143 – Can you clarify if adjusted relative risk values should be used for PAF calculation instead of relative risk?

For lines 198 to 201 – Can you give any rationale/reasoning/references for choosing the six explanatory variables that have been chosen?

Based on your results, specifically tables 1 and 2, can anything be commented/discussed on relationship between PAF and direct costs that you have observed?

In discussion, can you compare findings of high costs and mortality for males as compared to females with any existing evidence from Brazil to possibly validate your result?

For Table 4 from Line 290 onwards – I do not understand the unit used for your findings. The reported YLL is for what unit of observation? Is it for the entire cohort, per number of cases or what exactly?

In discussion section, can you compare your indirect cost findings to any other relevant literature?

7. PLOS authors have the option to publish the peer review history of their article (what does this mean?). If published, this will include your full peer review and any attached files.

Reviewer #1: No

Reviewer #2: No

---

## [Author Response · Author response to Decision Letter 1]

10 Nov 2022

9th Nov 2022

REBUTTAL LETTER

PONE-D-21-41019: The direct and indirect costs of cardiovascular diseases in Brazil

Dear Pankaj Bahuguna, Ph.D.

We would like to thank you and the reviewers for your time to provide feedback on our paper. We have addressed all Journal requirements and the reviewers’ comments below. 

Journal Requirements

Comments to the Author: 

1. “Please review your reference list to ensure that it is complete and correct. If you have cited papers that have been retracted, please include the rationale for doing so in the manuscript text, or remove these references and replace them with relevant current references. Any changes to the reference list should be mentioned in the rebuttal letter that accompanies your revised manuscript. If you need to cite a retracted article, indicate the article’s retracted status in the References list and also include a citation and full reference for the retraction notice.”

Response: Thank you for raising this point. We extensively reviewed the reference list and, in this manuscript, we haven’t cited papers retracted.

Reviewer #1

General comments to the Author: 

“Thank you for revising the manuscript and improving clarity for better understanding of the readers. However, the paper still needs language editing at few places and use of shorter sentences.”

Response: Thank you for your comments and for the opportunity to further improve our manuscript with your suggestions.

Specific comments to the Author: 

Abstract

Line 24-25: It could be useful to know whether the cost mentioned here was overall cost/health system cost/societal cost and specifically for treatment? 

Response: Thank you for your suggestion. We added the information that the costs were from health system and related to hospitalizations and outpatient care.

In Lines 26-31, now it says: 

“The calculation of direct costs for cardiovascular diseases (CVD) was made from the accounting sum of costs with hospitalizations and outpatient care found in the National Health System (Hospital Information System and Outpatient Information System), from 2017 to 2019, including the costs of treatment, such as medical consultations, medical procedures, and drugs. Regarding the indirect costs, they were measured by the loss of human capital, given the premature death, resulting in loss of productivity.”

Line 27-28: Were these direct medical costs only? Please specify if so. 

Response: Thank you for pointing out this omission. We added the costs’ information. 

In Lines 29-30, now it says:

“…including the costs of treatment, such as medical consultations, medical procedures, and drugs.” 

Line 29-30 can be rephrased to shorter sentences for better clarity.

Response: Thank you. We rephrased the sentence.

In Lines 30-31, now it says:

“Regarding the indirect costs, they were measured by the loss of human capital, given the premature death, resulting in loss of productivity.”

Results: Line 33: Higher burdens of what attributable to?

Response: Thank you. We now rephrased the sentence.

In Line 33, now it says: 

“Higher burden of CVD attributable to the consumption of salt,…”

Conclusion Line 39: “Studies of this nature….” Is very broad scope and could be rephrased to narrower scope for clarity. Also authors may use shorter sentences in conclusion and should restrict to findings from the study than generic statements.

Response: Thank you. We now used shorter sentences restricted to our findings.

In Lines 40-42, now it says: 

“The attributable fractions to consumption of salt are the ones that cause the most effects on CVD, followed by saturated fat and trans fats, with direct and indirect costs being higher for males.”

Introduction

Line 44: This line may be rephrased to “cardiovascular diseases (CVD) are one of the key challenges in health globally, since……………”

Response: Thank you for your suggestion. We rephrased the text using your suggestion.

Line 78: Please cite supporting references besides “few studies…….” Or may rephrase.

Response: Thank you for raising this point. Now we added two supporting references.

In Lines 78-79, now it says:

“Few studies have been carried out with the objective of attributing the effect of nutrient consumption to the cause of CVD and its consequent costs [23,24].”

Line 85-86 looks repetitive in lines 94-96 and may be better placed there.

Response: Thank you for spotting this. Now we deleted the lines 85-86 and placed only in Lines 100-102.

Lines 87-89 seems better suited to methodology section.

Response: Thank you for spotting this. Now we deleted Lines 87-89 from the Introduction and added to methods section (Lines 193-199).

Line 99 could be rephrased to “Several databases were referred to collect and analyse CVD cost related information between 2017 and 2019 for Brazil.”

Response: Thank you for your suggestion. We rephrased the text using yours (now in Lines 105-106).

Line 114: There seems to be incomplete word in here. Whether its three sources or three databases, it is currently unclear to understand “three bases”.

Response: Thank you for spotting this. We made it clearer in the text that we used three databases to obtain the information needed to calculate indirect costs.

In Line 121, now it says:

“Regarding indirect costs, three databases were necessary…”

Discussion 

“Line 293: The sentence could be rephased to “Findings from or study showed………….”

Response: Thank you for your suggestion. We rephrased the text using your suggestion.

Reviewer #2

Comments to the Author: 

“Thank you for considering suggestions and revising the manuscript. The manuscript content is structured better and reads more logically now. The authors have addressed grammatical mistakes to a large extent. Policy relevance of this work is highlighted better than the previous version. Following are some of my suggestions/clarifications for the revised manuscript (Line numbers refer to the latest unmarked version):”

Response: we are very grateful for all the considerations. They were very valuable to improve our manuscript.

“Consider further grammatical checks: For example, replace burdens with burden (multiple places, replace servings with serving – Line 158, typos on Line 214-215, Replace manuscript with ‘study’ – Line 293, typo error on Line 295, typo on Line 375, consider replacing ‘inadequate diet’ with maybe unhealthy diet if you thing that is more reflective on Line 305. You may consider further similar grammatical checks in the manuscript.” 

Response: Thank you for raising these points. We checked the whole manuscript looking for grammatical errors. Now the corrections are in red words.

“The revised introduction section makes a better read. However, on line 87, you go on to directly explain methods of indirect cost analysis. I would suggest you consider moving this to the methods section. The introduction section could give a brief overview of what all analysis (PAF, direct cost, indirect costs) is being aimed. Similarly, I found it difficult to establish relevance of Line 69-70 in context to the previous statement. What do you mean by recurrent costs here? My understanding is recurrent costs would be those attributed to hospitalisations and consultations (22% of costs)”

Response: Thank you for your comments. We moved the Line 87 and correlated aspects to methods section, and we added three paragraphs (Lines 85-99) with a brief overview of all analysis, as suggested. In Lines 69-70, we are talking about all kind of costs mentioned in previous statement (mortality, hospitalization and years of life lost). To make this information clearer, we have amended the text using “these recurrent costs”.

In Lines 85-99, now it says:

“The Population Attributable Fraction (PAF) is a measure of public health impact that has been widely used by the World Health Organization, based on Global Burden Disease (GBD) data, in order to determine goals, prioritize interventions and build public policies [25,26]. In addition, the PAF has also been used to provide information on economic costs attributable to some risk factors, such as salt consumption [27].

There are two ways to estimate health costs for a disease. The first is “top-down”, going from the total values at the national level of the set of all diseases and, through a disaggregation process, arriving at the level at which the cost of the disease under analysis is found. The second is “bottom-up”, and through this method, estimates are made for a sample of cases and are extrapolated to the total number of individuals [28]. 

In Brazil’s context, it is possible to obtain the total direct costs related to a given pathology in the National Health System, which can be disaggregated by level of health care (outpatient and hospital), sex and age groups. Thus, the best approach to be used in Brazil is the top-down approach, from the perspective of public health services, based on health cost data, available in the information systems of the Ministry of Health [28].”

In Lines 69-70, now it says:

“Therefore, the growth of these recurrent costs represents an important problem for health systems, as well as for their socioeconomic impacts.”

“For lines 137 to 143 – Can you clarify if adjusted relative risk values should be used for PAF calculation instead of relative risk?” 

Response: Thank you for raising this point. To calculate the relative risk (RR) used in the PAF formula, it was calculated according to the distribution of our own variables. For this, we used a relative risk already established in the literature, from review studies or larger studies, such as the Global Burden Disease (GBD). These relative risks are adjusted for, at least, sex and age. Implicit in the way we characterize the RR function are some of the fundamental assumptions we make about relative risk. That is, the relative risk increases exponentially as the distance from the theoretical minimum level of risk exposure (y) increases, that there is no risk associated with exposure beyond the theoretical minimum level of risk exposure, and that both x and the level theoretical minimum risk exposure for an individual at exposure level x are the qth quantile of their respective distributions (the observed exposure distribution and the TMREL, respectively).

Reference:

Micha R, Peñalvo JL, Cudhea F, Imamura F, Rehm CD, Mozaffarian D. Association Between Dietary Factors and Mortality From Heart Disease, Stroke, and Type 2 Diabetes in the United States. JAMA. 2017 Mar 7;317(9):912-924.

“For lines 198 to 201 – Can you give any rationale/reasoning/references for choosing the six explanatory variables that have been chosen?” 

Response: Thank you for your comments. The Mincerian equations from the study by Mincer (1974) show that the coefficients of age (proxy for experience), education and sex are significantly relevant to estimate individual income. For Brazil, due to regional/local and color/race disparities, the location variables "individual residence" and "color/race" are also of great importance. We added the reference in the text. 

In Lines 212-216, now it says:

“In this study, irreversible productivity losses due to the premature death of individuals aged 25 to 65 years caused by CVDs were considered. For this task, six explanatory variables for income that are available in the NHSS, and SIM databases were considered: state of the country in which the individual resides (residence), age, sex, education level, color/race, and marital status [39].” 

“Based on your results, specifically tables 1 and 2, can anything be commented/discussed on relationship between PAF and direct costs that you have observed?”

Response: Thank you for your advice. We added a paragraph in the discussion.

In Lines 323-328, now it says:

“Although the highest CVD costs attributable to the consumption of salt, saturated fat and trans fats is among the 45-74 age groups, the consumption of these nutrients is inversely proportional to the age. One of the explanations for our findings is that, due to the greater number of CVD cases occurring in these age groups, the risk attributable to the consumption of these nutrients is added to the risk related to the age group, suggesting that preventive measures at younger ages can benefit the entire population in their future.”

“In discussion, can you compare findings of high costs and mortality for males as compared to females with any existing evidence from Brazil to possibly validate your result?”

Response: Thank you for spotting this. In discussion, a study that addressed this issue with Brazilian data had already been mentioned. However, to reinforce our findings, we add another recent evidence.

In Lines 337-345, now it says:

“Nilson et al. (2021) [24] estimated the economic effects and impact on health between 2013 and 2032 of the implementation of sodium reduction in processed foods in Brazil. They also reported that during this period, around 110,000 CVD male cases and 70,000 CVD female cases will be prevented. This estimate will result in a total savings of around US $ 220 million in medical costs for the Brazilian National Health System for the treatment of CHD and stroke. Moreover, corroborating our findings, a recent study carried out in Brazil showed that the costs attributable to excessive salt consumption were higher for males when compared to females, corresponding to 62% of the costs associated with hospitalizations and 53% of outpatient’s costs for CVD, attributable to salt consumption [27].”

“For Table 4 from Line 290 onwards – I do not understand the unit used for your findings. The reported YLL is for what unit of observation? Is it for the entire cohort, per number of cases or what exactly?”

Response: Thank you for your question. The Years of Life Lost (YLL) for a cause are essentially calculated as the number of cause-specific deaths multiplied by a loss function, specifying the years lost for deaths as a function of the age at which death occurs. Table 4 shows the total years of life lost attributable to the consumption of nutrients (salt, saturated fat and trans fats) by age group and sex. The unit of measurement is actually "years", as presented in the Global Burden Disease (GBD) and as presented in other publications (please, see the doi https://doi.org/10.1371/journal.pone.0235514.t003, as example).

“In discussion section, can you compare your indirect cost findings to any other relevant literature?”

Response: Thank you for your suggestion. The problem with making comparisons in relation to indirect costs lies precisely in the difference between the methodologies adopted in studies involving this theme. Recent work by Nilson et al (2020), assessed the years of life lost, however, it did not calculate indirect costs. Rasmussen's study (2015), for example, which carried out an analysis of indirect costs, addressed other methodological aspects, such as absenteeism, presenteeism and early retirement. So, we decided to add a few words to the discussion referring to a limitation of comparing these data given these previous explanations. 

In Lines 454-455, now it says:

“Besides, the fact that other studies use different approaches to calculate the indirect costs makes it difficult to compare the results [12,64,65].”

Patrícia Moreira

On Behalf of co-authors

---

## [Editor Report · Decision Letter 2]

24 Nov 2022

The direct and indirect costs of cardiovascular diseases in Brazil

PONE-D-21-41019R2

Dear Dr Moreira,

We’re pleased to inform you that your manuscript has been judged scientifically suitable for publication and will be formally accepted for publication once it meets all outstanding technical requirements.

Kind regards,

Pankaj Bahuguna, Ph.D.

Guest Editor

PLOS ONE
---

## [Editor Report · Acceptance letter]

14 Dec 2022

PONE-D-21-41019R2 

The direct and indirect costs of cardiovascular diseases in Brazil 

Dear Dr. Moreira:

I'm pleased to inform you that your manuscript has been deemed suitable for publication in PLOS ONE. Congratulations! Your manuscript is now with our production department. 

Kind regards, 

on behalf of

Dr Pankaj Bahuguna 

Guest Editor

PLOS ONE